# Leishmaniasis in deployed military populations: A systematic review and meta-analysis

Ngwa Niba Rawlings[1,2]\*, Mark Bailey[3,4], Orin Courtenay[2,5]

**1** Department of Environmental Health, Defence Medical Services, Ministry of Defence, London, United Kingdom, **2** School of Life Sciences, University of Warwick, Warwick, United Kingdom, **3** Warwick Medical School, University of Warwick, Warwick, United Kingdom, **4** Department of Military Medicine, Royal Centre for Defence Medicine, Birmingham, United Kingdom, **5** Zeeman Institute, University of Warwick, Coventry, United Kingdom

\* ngwa.niba-rawlings@warwick.ac.uk

## Abstract

Leishmaniasis affects military personnel deployed to endemic areas following exposure to sand flies infected with the protozoa *Leishmania*. This systematic review and meta-analysis of data specific to military populations aims to identify knowledge gaps to mitigate sand fly exposure and *Leishmania* transmission during deployments. The review was registered on PROSPERO (CRD42023463687). Random-effects meta-analyses and narrative synthesis were performed. Thirty-six studies were included, most of which reported on cutaneous leishmaniasis (CL), showing a mean cumulative incidence of 10% (95% CI: 5–16), suggesting higher rates in the Eastern Mediterranean region (14% [95% CI: 12–16]) compared to the African region (8%) and American region (9%). Asymptomatic *Leishmania* infection had a cumulative incidence of 11% (95% CI: 6–17), with higher rates in Eastern Mediterranean countries (20% [95% CI: 14–25]). Diagnosis involved parasitological, serological, and molecular methods, with *L. (L) mexicana* and *L. (V.) braziliensis* identified as the predominant CL pathogens in deployed troops in the Americas. Visceral leishmaniasis cases were less frequent, all reported from the Eastern Mediterranean and associated with the *Leishmania donovani/infantum complex*; whereas CL cases in the Old World were due predominantly to *L. major* and *L. tropica*. Regular use of long-lasting insecticidal nets to mitigate sand fly exposure demonstrated high potential effectiveness than other reported personal protective measures (PPMs) which yielded mixed or inconclusive results. In summary, the systematic review revealed the substantial variability between study designs and statistical integrity. There is need for more consistent and robustly designed studies including well-define controls and replication. Future studies would be advised to explore the long-term effectiveness and practicality of PPMs, both individually and in combination, across diverse deployment settings.

**Data availability statement:** All relevant data are in the manuscript and its supporting information files.

**Funding:** This work was supported by the Defence Medical Services (DMS) Research and Clinical Innovation (RCI) (RE: 22/23.003) and the Drummond Foundation (Horne/211122) (Both to NNR) The funders had no role in study design, data collection and analysis, decision to publish, or preparation of the manuscript.

**Competing interests:** The authors have declared that no competing interests exist.

## Author summary

Leishmaniasis is a parasitic disease transmitted by sand flies that affects populations worldwide, with particularly increasing incidence among military personnel deployed to endemic regions. This systematic review and meta-analysis focuses on leishmaniasis among military populations, examining rates of infection, regional variations, and the effectiveness of preventive measures. Our study synthesises data from thirty-six studies, revealing a 10% overall incidence of cutaneous leishmaniasis (CL) among deployed troops, with the highest rates found in the Eastern Mediterranean region. Additionally, asymptomatic infections were observed at a cumulative rate of 11%, with significant regional variations. The review identifies *L. (L) mexicana* and *L. (V.) braziliensis* as the primary CL pathogens in the Americas, while *L. major* and *L. tropica* were the most common in the Eastern Mediterranean. The analysis also highlights the effectiveness of long-lasting insecticidal nets in reducing sand fly exposure, though other personal protective measures (PPMs) showed mixed results. Despite these findings, our study notes substantial variability in study design and statistical rigor across the included research, highlighting the need for more standardised, well-controlled studies with reproducible methodologies. This study provides recommendations for future research on leishmaniasis in military settings and beyond, advocating for long-term evaluations of PPM efficacy across diverse regions. The findings emphasise the importance of region-specific disease control strategies and add to the growing body of evidence guiding prevention, treatment, and policy in the neglected tropical disease community.

## Introduction

Leishmaniasis is a vector-borne parasitic disease of humans caused by over 20 different species of *Leishmania* (Kinetoplastida: Trypanosomatidae) transmitted by hematophagous female Phlebotomine sand flies (Diptera: Psychodidae) [1]. The disease is endemic in over 90 countries worldwide, with an estimated annual incidence of 700,000 to 1 million new cases globally [1–3]. The polymorphic outcomes of symptomatic (clinical) infection predominantly include cutaneous leishmaniasis (CL), mucocutaneous leishmaniasis (MCL), visceral leishmaniasis (VL, or kala-azar), and post-kala-azar dermal leishmaniasis (PKDL) [1]. CL is the most common form, presenting as simple to severe cutaneous ulcers and disfigurement which can progress to destruction of the oral-pharyngeal mucosa (MCL) or diffuse CL [4,5] resulting in significant morbidity. VL is the most severe form of leishmaniasis causing fever, weight loss, spleen and liver enlargement, with a fatality rate of >95% without effective diagnosis and treatment [1].

Some 30 species of sand fly are proven or suspected vectors of *Leishmania* [6,7]. Sand fly blood-seeking behaviour is typically crepuscular/ nocturnal, though they may bite during daylight hours when disturbed [8]. A wide variety of vertebrates are indicated as sink (dead-end) or source (reservoir) hosts, the latter including humans, domestic dogs, lagomorphs, and synanthropic and wild rodents, depending on the *Leishmania* species-specific transmission cycle [9].

Military personnel are often exposed to infectious sand fly bites when deployed to endemic settings, notably in the Middle East [10–12], Africa [13], Europe [14,15] and the Americas [16–21], where the cumulative incidence of CL for example ranges from 0.2% to 25.2% [3,10–12,14–23]. VL is less common amongst the military [22,24–26], accounting for, e.g., 1.2% of all leishmaniasis cases reported between 2001 and 2006 among US soldiers deployed to Iraq, Afghanistan, Oman and Saudi Arabia [22]. In the British military, only one VL case has been recorded to date, and which followed deployment to Iraq [MoD restricted data].

Development of leishmaniasis has significant health and financial consequences to both afflicted individuals and to military institutions. Patients suffer short to long-term health conditions resulting in prolonged time out of service; chemotherapeutic treatments are costly and often toxic [27]; and financial compensation claims against the MoD may be extensive [23]. Currently, there are no chemoprophylaxis or human vaccines to protect humans against leishmaniasis [28], thus prevention primarily relies on mitigating against sand fly bites, on the control of the sand fly vector(s), and/or the mammalian reservoir populations [10].

Deployed personnel are usually supplied with personal protective measures (PPMs) accompanied by instructions of their use to best reduce exposure to sand fly bites. However, supply and compliant use of PPMs are not consistent, factors often associated with an increased risk of contracting leishmaniasis [16,17,20,21,29–34]. Previous reviews of leishmaniasis epidemiology in civilian populations do not address the specific circumstances of operational deployments [35–39]. This systematic review and meta-analysis of data specific to military populations aims to help identify knowledge gaps towards mitigating exposure to sand fly vectors and *Leishmania* infection during deployment.

## Methods

The review is registered on PROSPERO (CRD42023463687). The Preferred Reporting Items for Systematic Review and Meta-Analysis (PRISMA) guidelines [40,41] and the Meta-analysis Of Observational Studies in Epidemiology (MOOSE) guidelines [42] were followed (Tables I and J in S1 Text).

### Search strategy and selection criteria

A comprehensive literature search was developed collaboratively with the review team through an iterative process to identify peer-reviewed studies reporting any form of leishmaniasis among military personnel deployed to any location. The search was performed in multiple databases, including MEDLINE, EMBASE, Web of Science, Scopus, and ProQuest. Hand searches were also conducted in preprint servers (bioRxiv and medRxiv), article reference lists, and Google Scholar Citations for additional relevant studies.

The search strategy combined MeSH terms and keywords, including terms related to sand fly, leishmaniasis, and military. Truncation and wildcards were used to capture variations of words such as 'leishman*' (to capture leishmaniasis, leishmanial, *Leishmania*). Specific subtopics or aspects of leishmaniasis in the military was considered such as cumulative incidence, incidence proportion, attack rate, sand fly and *Leishmania* species characterisation, sand fly saliva antigen exposure and risk factors. The search was limited to studies published in English (or with translations into English) between January 1990 and October 2023.

Inclusion criteria were studies of deployed military populations reporting any of the following: cumulative incidence (incidence proportion or attack rate) of leishmaniasis, characterisation of sand fly and *Leishmania* species, sand fly saliva exposure, the effectiveness of PPMs and associated risk of developing laboratory confirmed asymptomatic or symptomatic leishmaniasis. Studies reporting cases of leishmaniasis without laboratory confirmation were excluded. Studies of non-military populations were also excluded. The screening results are reported following the PRISMA statement [40].

### Data extraction and quality assessment

Two authors independently conducted screening of titles and abstracts, review of full texts, data extraction, and performed quality assessment (rating) using the National Institutes of Health Quality Assessment Tools [43], and the Newcastle Ottawa scale [44] (Tables B and C

in S1 Text). To ensure accuracy, the screening results were manually cross-checked, and any disagreements in data extraction or quality assessment were resolved through consensus. Data extraction included characteristics of the study design, setting, numbers and characteristics of confirmed subclinical and clinical leishmaniasis, study population size, and period of deployment to the endemic setting. Information on human immune responses to sand fly saliva proteins exposure (as a measure of individual sand fly bite exposure), and characterisation of the sand flies, the infecting *Leishmania* species and methods of detection, were also extracted (Table A in S1 Text).

## Data analysis

The geographical locations of military deployments were grouped by WHO-defined geographical region [45]. The extracted data on the numbers of cases and the number of troops deployed over the reported deployment period were used to calculate the cumulative incidence [46] otherwise known as the incidence proportion, as the number of new asymptomatic or symptomatic confirmed cases divided by the total number of troops at the start of the deployment period. Deployment periods were short (median 1 months, maximum 6 months), and we assumed that the deployed troops had no previous history of leishmaniasis, were fully susceptible to infection, and there was no significant loss-to-follow-up. PPMs and associated risk of developing leishmaniasis were assessed for their suitability to be pooled in a meta-analysis. Effect sizes for single proportions (cumulative incidence), and precomputed effect sizes for the effectiveness of PPMs and associated risk of developing leishmaniasis, were calculated and declared. Potential impacts of data transformations on effect sizes (Freeman-Tukey, and Logit) were compared to the raw proportions, confirming no substantial differences in the overall estimates (Figs F–I in S1 Text). Thus, the Freeman-Tukey method for estimating cumulative incidence was adopted and reported here. For all meta-analyses, data were fitted to random effects models using restricted maximum likelihood [47,48]. For evaluation of the effectiveness of PPMs and associated risk of developing leishmaniasis, the precalculated effect sizes (odds ratios [OR] and confidence intervals) reported in the identified publications were used to calculate the natural log OR (ln[OR]) and associated standard error (SE), after which they were pooled for meta-analysis.

The degree of heterogeneity between study outcomes was assessed using $I^2$ (I-squared), $\tau^2$ (Tau-squared), $H^2$ (H-squared), and Q (Cochran's Q test) statistics. Low values of $\tau^2$ (near 0), $I^2$ (close to 0%), $H^2$ (near 1), and Q with p-values $p > 0.05$, suggested low or no heterogeneity, while higher values of $\tau^2$, $I^2$ >75%, $H^2$ >1, and Q with a low p-value $p < 0.05$, indicated substantial heterogeneity [49,50]. The Z-test was used to assess whether the overall pooled effect size was significantly different from zero. Publication bias was assessed using the Egger's test [51]. For meta-analysis including at least five studies, sensitivity analyses were performed, showing changes in the summary effect estimate and confidence intervals resulting from excluding one study at a time from the meta-analysis (Tables E–H in S1 Text). As meta-analysis was not possible to address all the objectives of this review due to quantitative data limitations, a narrative synthesis was conducted following guidelines of Popay et al. [52].

## Results

The literature search yielded a total of 1,456 studies (Fig 1). Following the removal of duplicates, 610 studies underwent screening based on their titles and abstracts. Of these, 468 records were excluded because they were not relevant to the subject of this review. The full text of 142 studies were assessed for eligibility, of which 36 met the inclusion criteria (Fig 1). Of the nine excluded non-English publications without an English translation, only two were potentially relevant.

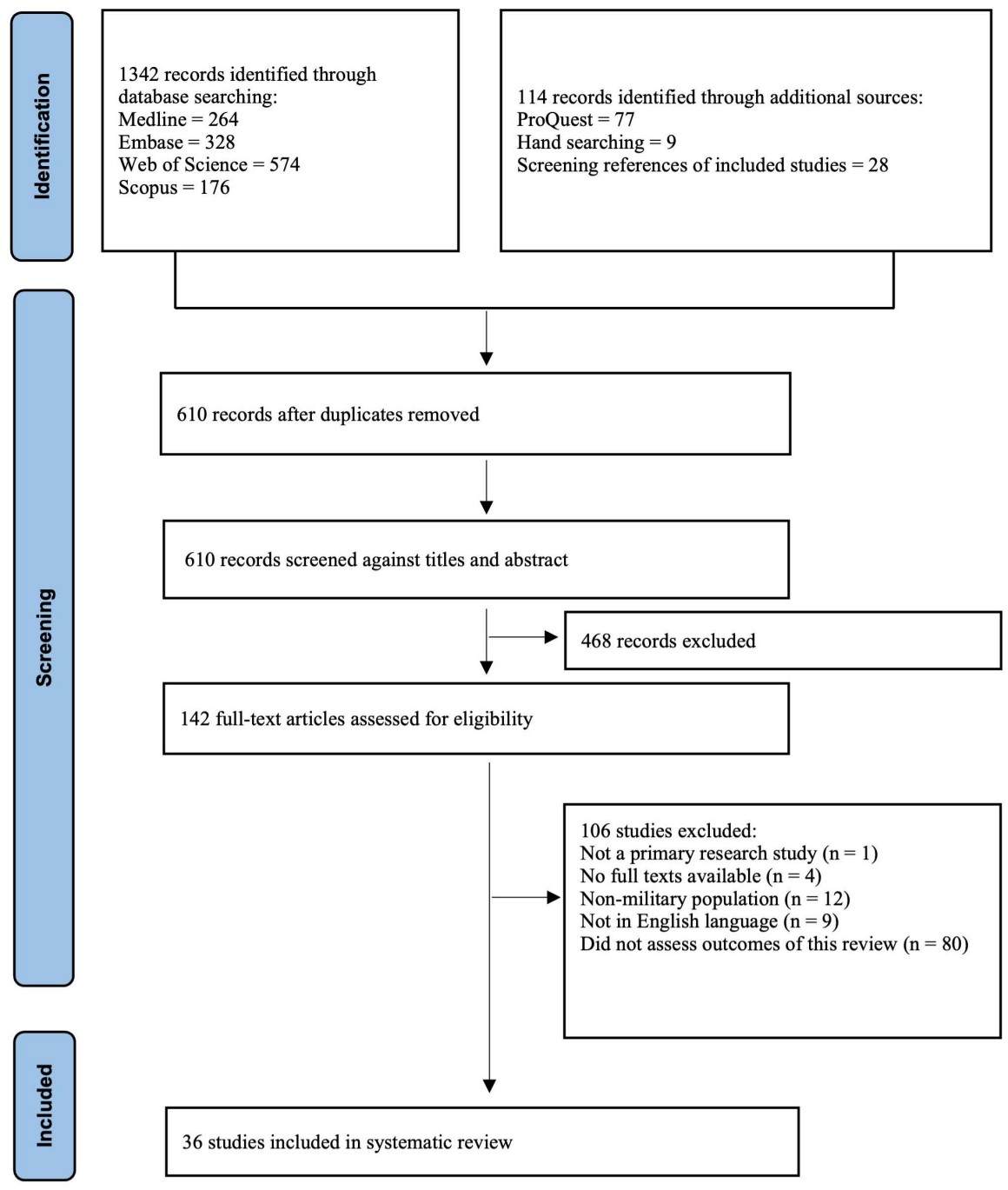

**Fig 1. PRISMA flow-chart showing process of study selection.**

## Description of included studies

The 36 included studies reported on military personnel deployed to the American region (AMR; 16 studies), the Eastern Mediterranean region (EMR; 15 studies), the South-East Asian region (SEAR; 2 studies) and 1 study each deployed to the African region (AFR), the European region (EUR), and to countries spanning two regions (EUR and EMR) (Table A in

S1 Text). Out of the 36 studies, 8 (22.2%) reported the cumulative incidence of symptomatic leishmaniasis, 3 (8.3%) the cumulative incidence of asymptomatic leishmaniasis, 24 (66.7%) characterised the *Leishmania* species aetiology, 7 (19.4%) identified the local sand fly species, 4 (11.1%) reported on soldier immune responses to sand fly salivary proteins, and 13 (36.1%) reported on the effectiveness of PPMs and associated risk of developing subclinical and clinical leishmaniasis. Of the 24 studies that reported confirmed clinical outcomes, 20 (83.3%) studies reported CL, three (12.5%) reported VL, and one study (4.2%) reported both CL and VL (deployment to Afghanistan) [53].

The 36 study designs comprised six cohort studies [17,21,33,34,54,55], fifteen cross-sectional studies [14,15,20,27,30,32,56–64], one mixture of cohort and case-control study [16], two mixtures of cross-sectional and cohort study [13,65], four case-control studies [66–69], three case reports [24,53,70] and five case series [25,31,71–73]. The publication dates of the studies ranged from 1992 to 2023, with the majority (*n* = 20, 57.1%) published between 2011 and 2023. The number of participants per study ranged from 1 to 5000 soldiers. Study characteristics (location, study design and sample size) and quality ratings, as defined in Methods, are shown in Tables A-C in S1 Text. Eleven studies reported cumulative incidence, eight of symptomatic CL [13,16,17,20,21,33,34,71] and three of asymptomatic (subclinical) confirmed *Leishmania* infections [14,15,69]. Thirteen cases of symptomatic VL were reported in three studies [24,25,73] which were too few to perform a meta-analysis, hence are described under the narrative accounts below.

## Meta-analyses

**Cumulative incidence of symptomatic cutaneous leishmaniasis.** Of the eight studies that reported the cumulative incidence of symptomatic CL (eleven estimates), five studies were among soldiers deployed to AMR [16,17,20,21,71], two to EMR [33,34] and one to AFR [13]. The duration of deployment ranged from <1 month [20,71], 1 – 4 months [13,17,21] to >4 months (maximum 6 months) [16,33,34]. Meta-analysis of the pooled Freeman-Tukey transformed proportions estimated an overall cumulative incidence of symptomatic CL of 10% (95% C.I.: 5, 16) with significant heterogeneity ($I^2$ = 97.84%) between studies (Fig 2), and no evidence of publication bias ($z$ = 1.85, $p$ = 0.065) (Fig A in S1 Text ). The seven estimates from the five studies within AMR showed similar heterogeneity ($I^2$ = 97.90%) (Fig D in S1 Text ).

High cumulative incidence of symptomatic CL was in troops deployed to AMR [17,21], though not exclusively so (Figs 2 and D in S1 Text ). Pooled estimates by region suggested highest values in the EMR (14% [95% CI: 12, 16]), though no statistical difference was detected between WHO regions (Bayesian ANOVA [$Q_b$] = 1.90; $P$ = 0.39). The cumulative incidence associated with wet *versus* dry season of deployment was not dissimilar (10% [95% CI: 5, 16]) (Fig 3). The duration of deployment lasting 1-4 months appeared to result in a higher cumulative incidence (15% [95% CI: 07, 25]) compared to <1 month and >4 months deployments (<9%) (Figs 3 and E in S1 Text ).

**Cumulative incidence of asymptomatic *Leishmania* infection.** Asymptomatic *Leishmania* infections were detected in troops post-deployment by testing for anti-*Leishmania* antibodies using enzyme-linked immunosorbent assay (ELISA) [14], or by both ELISA and screening for *Leishmania* DNA by polymerase chain reaction (PCR) [15,69]. Of the three identified studies, one reported on troops deployed to EMR [69], one to EUR [14] and one deployed to both EMR and EUR [15].

Two molecular studies detected *L. donovani/infantum complex* DNA in blood samples. One study of Austrian troops deployed to Syria (EMR), Lebanon (EMR), and Bosnia and

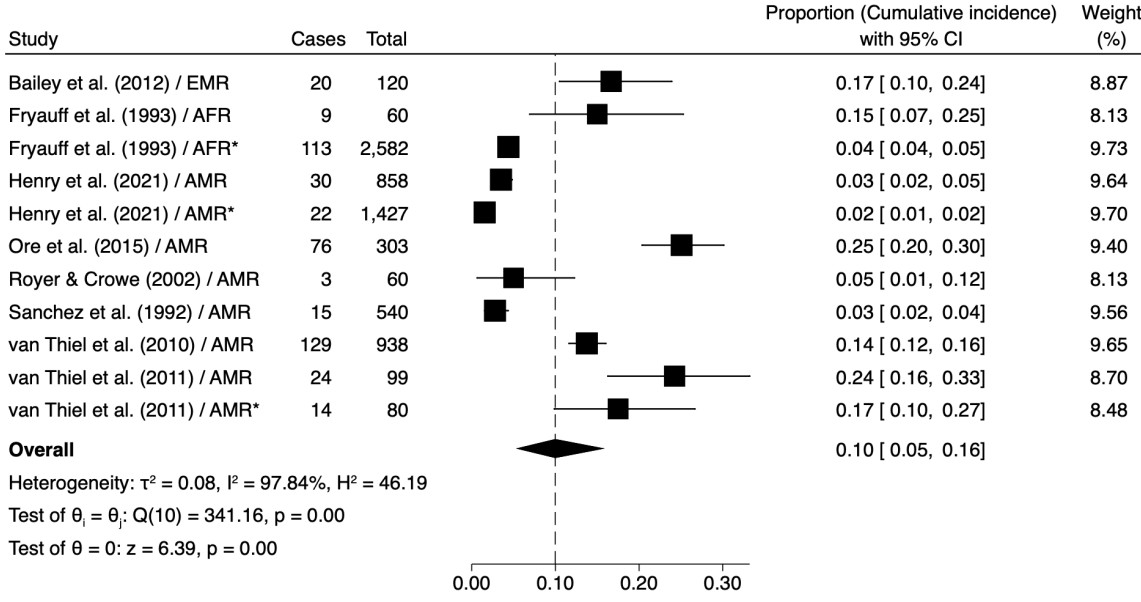

**Fig 2. Meta-analysis of the pooled estimate of the cumulative incidence of symptomatic CL in deployed military personnel.**
*cumulative incidence data reported for different subpopulations in the same study. 95% C.I. confidence interval. Weight (%), relative influence on the overall estimates. $\tau^2$ - Tau-squared, $I^2$ - I-squared, $H^2$ - H-squared, Q - Cochran's Q test and Z - Z-test. **References:** Bailey et al. (2012) [33], Fryauff et al. (1993) [13], Henry et al. (2021) [16], Ore et al. (2015) [21], Royer & Crowe (2002) [71], Sanchez et al. (1992) [20], van Thiel et al. (2010) [34] and van Thiel et al. (2011) [17].

Herzegovina (BIH-EUR), identified by applying *Leishmania* Internal Transcribed Spacer Reverse (LITSR) and *Leishmania* 5.8S ribosomal RNA (L5.8S) primer pairs [15]. The other study of US troops deployed to Iraq (EMR) identified based on TaqMan specific primers [69]. The third study detected anti-*Leishmania* antibodies using RIDASCREEN *Leishmania* AB ELISA in Austrian troops deployed to Kosovo [14].

Meta-analysis of the Freeman-Tukey transformed proportions indicated an overall cumulative asymptomatic *Leishmania* infection incidence of 11% (95% CI: 6, 17) (Fig 4). The highest cumulative incidence value was for Iraq (EMR) (20% [95% CI: 14, 25]), with significant inter-study heterogeneity ($I^2$ = 78.47%) (Fig 4), and evidence of publication bias ($z$ = -2.64, $p$ <0.001) (Fig B in S1 Text).

**Adoption of PPMs and associated risk of developing leishmaniasis.** Thirteen studies reported on the effectiveness of PPMs and associated risk of developing leishmaniasis, of which eight studies could be pooled for meta-analysis [16,21,30,32–34,56,69]. The majority of the studies reported risk factors for developing symptomatic CL. Considering the combined influences of up to six reported PPMs, the odds of leishmaniasis was reduced by 35% (OR = 0.65; 95% CI = 0.42, 0.89; $P$ <0.001), with moderate heterogeneity ($I^2$ = 66.86%) between studies (Fig 5). The regular use of military-issued Long-Lasting Insecticidal Nets (LLINs) indicates a 49% reduction in the odds of leishmaniasis, with a statistically significant protective effect and no heterogeneity (OR = 0.51; 95% CI = 0.24, 0.77; $P$ <0.001) between the four studies. The regular use of long-sleeved clothing (OR = 0.72; 95% CI = 0.26, 1.17; $P$ <0.001 $I^2$ = 74.41%]), insect repellent (OR = 0.40; 95% CI = -0.23, 1.03; $P$ = 0.21 $I^2$ = 81.79%), and insecticide treated uniforms (ITUs) (OR = 0.22; 95% CI = -0.36, 0.80; $P$ = 0.46 $I^2$ = 0.00%) may reduce the odds of leishmaniasis by 28%, 60% and 78%, respectively, however the latter two PPMs were not individually significantly different from controls (Fig 5). Knowledge about

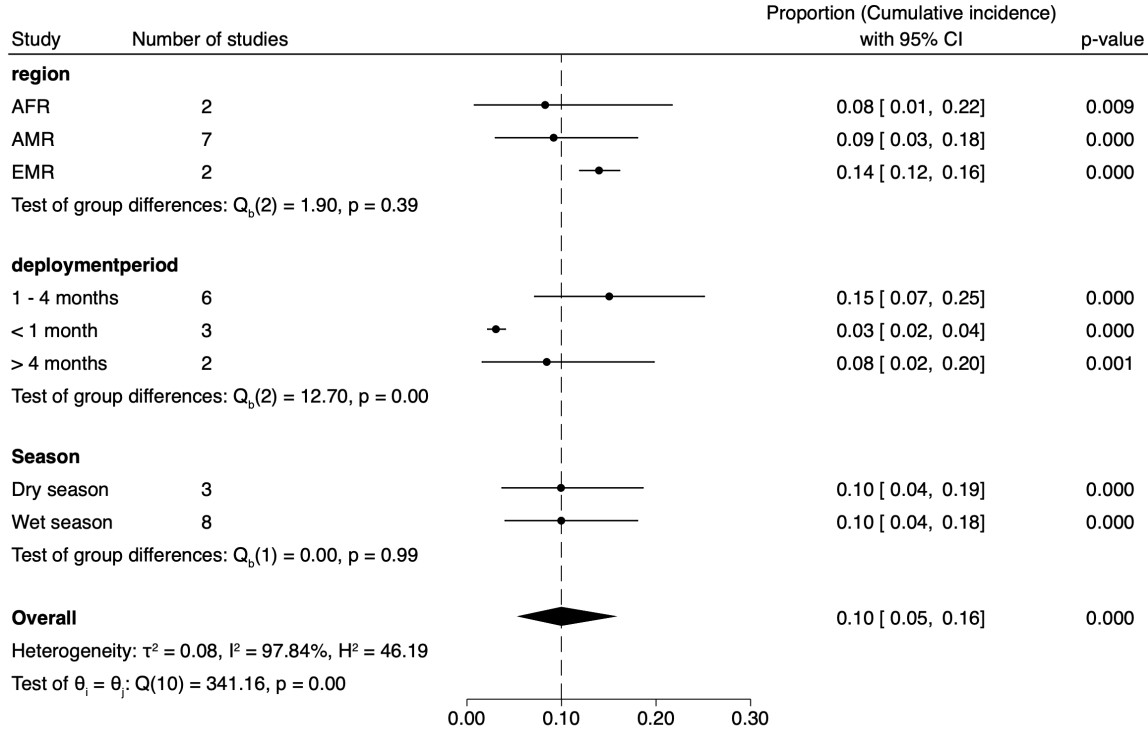

**Fig 3. Subgroup analysis of the cumulative incidence of symptomatic CL in deployed military personnel.** In this meta-analysis, there is no baseline of comparison, such as a line of no effect. Instead, the data is presented relative to an overall cumulative incidence line, which represents the combined or overall incidence across the studies analysed. 95% C.I. confidence interval. $\tau^2$ - Tau-squared, $I^2$ - I-squared, $H^2$ - H-squared, Q - Cochran's and Z - Z-test.

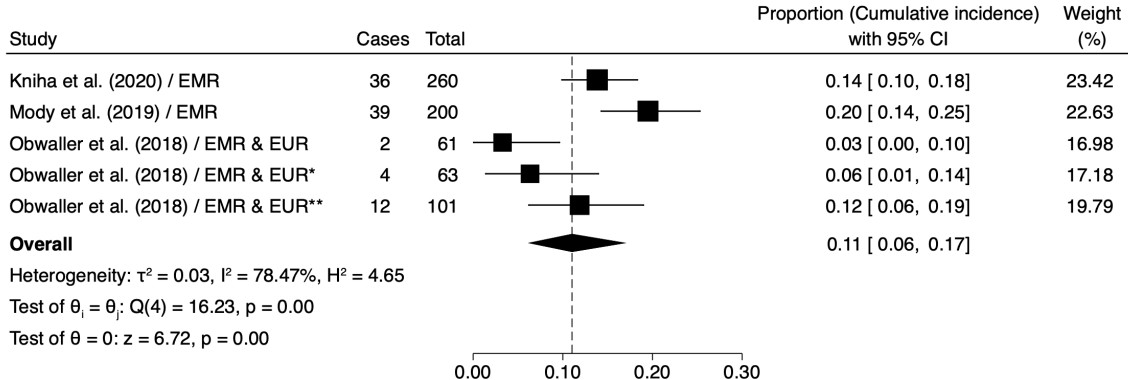

**Fig 4. Meta-analysis of the pooled estimate of the cumulative incidence of asymptomatic leishmania infection in deployed military personnel.** *cumulative incidence reported for different subpopulations in the same study. 95% C.I. confidence interval. Weight (%), relative influence on the overall estimates. $\tau^2$ or Tau-squared, $I^2$ or I-squared, $H^2$ or H-squared, Q or Cochran's Q test and Z or Z-test. **References:** Kniha et al. (2020) [14], Mody et al. (2019) [69], Obwaller et al. (2018) [15].

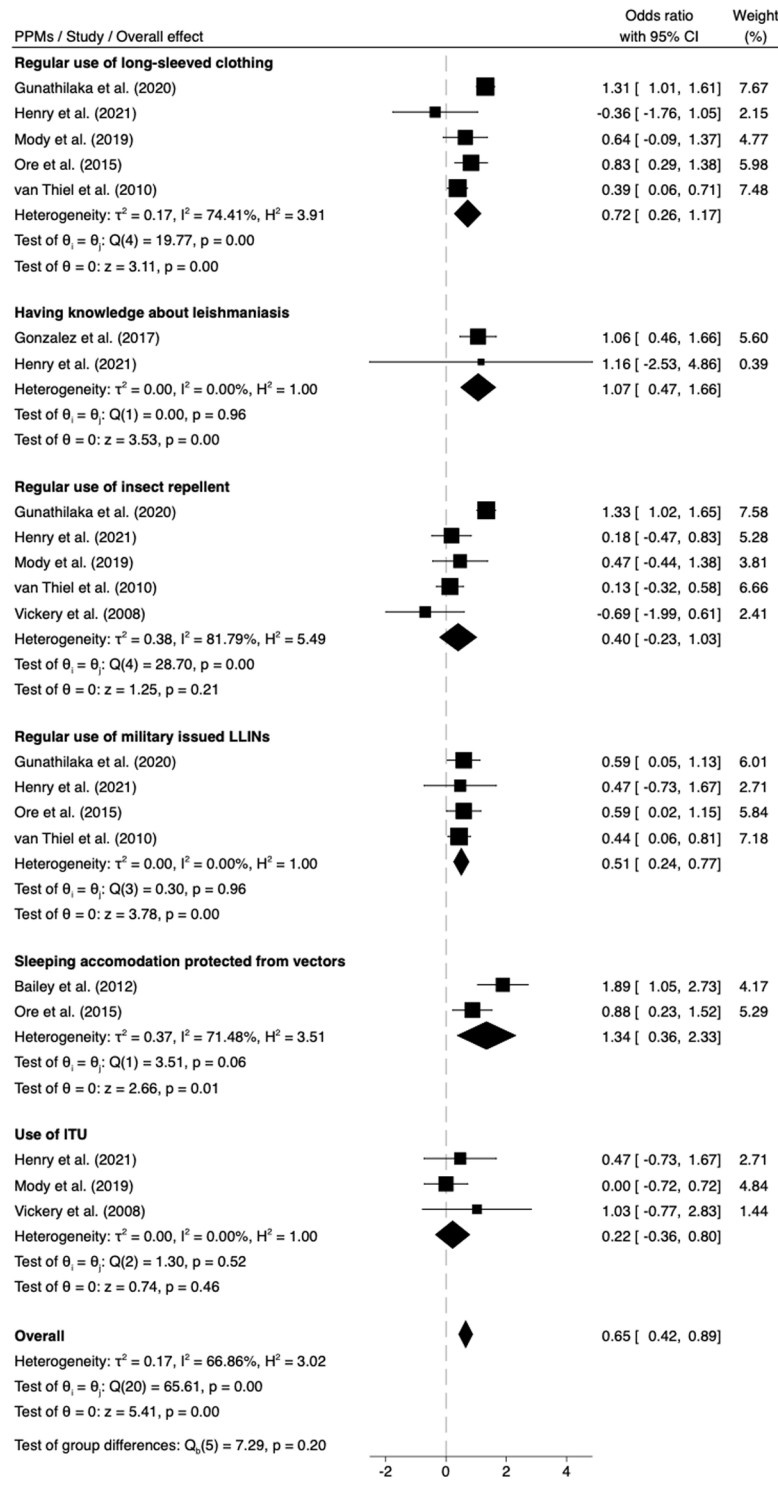

**Fig 5. Meta-analysis showing the association of six personal preventive measures (PPMs) with the reported odds ratio (OR) of leishmaniasis in deployed military personnel.** LLINs long-lasting insecticidal nets; ITUs insecticide treated uniforms. $\tau^2$ Tau-squared, $I^2$ I-squared, $H^2$ H-squared, Q Cochran's Q test. **References:** Bailey et al. (2012) [13,33], Henry et al. (2021) [16], Ore et al. (2015) [21], van Thiel et al. (2010) [34], Mody et al. (2019) [69], Gunathilaka et al. (2020) [32], Gonzalez et al. (2017) [56] and Vickery et al. (2008) [30].

leishmaniasis (OR = 1.07; 95% CI = 0.47, 1.66; *P* <0.001 *I²* = 0.00%) and sleeping in protected accommodation (OR = 1.34; 95% CI = 0.36, 2.33; *P* = 0.01 *I²* = 71.48%), were both significantly associated with reduced odds of leishmaniasis (Fig 5). Funnel plots and Egger's test showed no evidence of publication bias for these evaluated PPMs (Fig C and Table L in S1 Text ).

**Heterogeneity, publication bias, and sensitivity analysis.** Significant heterogeneity (*I²* > 75%) was detected in some meta-analyses (Figs 3–5); however, as there were few studies, further analyses were not performed to identify differentiating factors. The Sensitivity analyses performed for the above analyses by excluding individual studies with replacement did not identify particular studies that significantly influenced the overall direction of association, effect size, statistical significance, or level of heterogeneity (Tables E–H in S1 Text ). Egger's test results indicated evidence of small-study effects and publication bias in the analyses of asymptomatic *Leishmania* infections (*z* = -2.64, *p* <0.001), but not in analyses of symptomatic CL (*z* = 1.85, *p* = 0.065), or the effectiveness of PPMs and associated risk of developing CL (Figs A-C and Table L in S1 Text ).

## Narrative synthesis

A narrative synthesis of symptomatic VL is provided as quantitative data to conduct a meta-analysis were not reported. Descriptions of *Leishmania* and sand fly species identifications, and troop immune responses to sand fly salivary antigen exposure, are also provided below.

**Symptomatic VL.** Thirteen cases of VL were described, all among US soldiers; eight cases deployed during Operation Desert Storm in Saudi Arabia [73], and five cases deployed during Operation Iraqi Freedom and Operation Enduring Freedom in Iraq, Afghanistan and Oman [24,25]. The eight cases from Saudi Arabia exhibited clinical symptoms of viscerotropic leishmaniasis which is a mild form of VL and usually associated with *L. tropica* infections. Of these eight cases, seven exhibited symptoms of varying degrees including unexplained fever, chronic fatigue, malaise, cough, intermittent diarrhoea, abdominal pain, adenopathy, and mild transient hepato-splenomegaly, whereas none of the soldiers presented cutaneous manifestations [73]. Six of the eight cases were confirmed *L. tropica* infections.

The five cases deployed to Iraq, Afghanistan and Oman all presented classical signs of VL including fever, night sweats, cough, abdominal pain, diarrhoea, anorexia, nausea, cytopenias, elevated liver-associated enzymes, splenomegaly, hepatomegaly and weight loss [24,25]. *Leishmania* amastigotes were visualised in bone marrow biopsies from two cases from Iraq and in liver biopsies from two cases from Afghanistan [25]. The single case from Oman was confirmed as infected with a member of the *L. infantum/donovani* complex[24].

***Leishmania* aetiology.** Eighteen studies reported the specific identification of *Leishmania* species using molecular techniques: eight in AMR [16,17,21,27,31,59,65,70]), seven in EMR [25,33,34,53,69,72,73], one in EUR [15], and two in SEAR [62,64]. The majority (15/18) of identifications were from symptomatic CL patients [16,17,21,27,31,33,34,53,59,62,64,65,69,70,72], two from VL [25,73], and one from an asymptomatic infected patient [14].

PCR was the most used nucleic acid amplification method, including conventional PCR (cPCR), real-time PCR (qPCR), nested PCR, and kinetoplast DNA (kDNA) PCR; nucleic acid sequence-based amplification (NASBA) was used in addition. The PCR assays targeted specific genes such as 5.8S rRNA, heat shock protein (HSP70), cytochrome b (Cytb), small subunit rRNA (ssrRNA), internal transcribed spacer (ITS1 and ITS2), and mini-exon. The identified *Leishmania* species from infected soldier samples are summarised in Table 1.

**Sand fly species identification.** Seven studies reported the identification of sand fly species captured in the deployment foci [13,57,58,60,61,63,65]. Four reports were from AMR

Table 1. *Leishmania* and sand fly species reported per country/region of troop deployment.

| Deployment details | | | | CL cases/Total troops deployed (Cumulative incidence) | *Leishmania* species identified in cases | Type of leishmaniasis & activity performed by military during outbreak | Sand fly species identified | *Leishmania* species identified in sand flies |
| Region | Country of deployment & Nationality of infected soldiers | Duration (Months) | Season[a] | | | | | |
| --- | --- | --- | --- | --- | --- | --- | --- | --- |
| AMR | Belize Dutch soldiers [17] | 1 – 4 | Wet | 14/80 (17%) [17] | *L. (V) braziliensis* and *L. mexicana* | CL Jungle training that involved sleeping on the ground in sleeping bags ('hard drills') | NSF | NSF |
| | British soldiers [54,55] | | | 24/99 (24%) [17] | *L. (V) braziliensis* and *L. mexicana* | | | |
| | Brazil [60,63,65] Brazilian soldiers | NR | NR | 40/NR (NR) | *L. (V) braziliensis* | CL Military training camp where nocturnal activities were conducted in tropical forests (Atlantic rainforest) including jungle warfare training | *Lutzomyia. complexa*[b]*, Lu. choti*[*], *Lu. longispina, Lu. amazonensis, Lu. sordelli, Lu. evandroi, Lu. walkeri, Lu. capixaba, Lu. naftalekatzi, Lu. schreiberi, Lu. braziliensis, Lu. quinquefer, Lu. tupynambai, Lu. whitmani, Lu. barrettoi, Lu. servulolimai, Lu. aragaoi, Lu. ayrozai, Lu. claustrei, Lu. furcata, Lu. migonei, Lu. oswaldoi, Lu. shannoni complex, Lu. wellcomei, Lu. umbratilis, Lu. viannamartinsi, Lu. yuilli pajoti, Lu. ruii, Lu. anduzei, Lu. olmeca nociva, Lu. georgii, Lu. squamiventris squamiventris, Lu. monstruosa, Lu. flaviscutellata, Lu. rorotaensis, Lu. trichopyga, Lu. davisi, Lu. ubiquitalis, Lu. paraensis, Lu. nematoducta, Lu. inpai, Lu. trispinosa, Lu. geniculata, Lu. corossoniensis, Lu. williamsi, Lu. hirsuta, Lu. tarapacaensis, Lu. pacae, Lu. sericea, Lu. spathotrichia, Lu. dendrophyla, Lu. tuberculata, Lu. pilosa, Lu. antunesi, Lu. inflata, Lu. ratcliffei, Lu. dreisbachi, Lu. damascenoi, Lu. triacantha, Lu. cuzquena, Lu. saulensis, Lu. abunaensis, Lu. bispinosa, Lu. bourrouli, Lu. eurypyga, Lu. pennyi, Lu. sp. of Baduel, Lu. cultellata, Lu. scaffi, Brumptomyia pintoi, Lu. bursiformis, Lu. abonnenci & Lu. lutziana* | *L. (V.) braziliensis* |

(Continued)

Table 1. (Continued)

| Deployment details | | Duration (Months) | Season^a | CL cases/Total troops deployed (Cumulative incidence) | Leishmania species identified in cases | Type of leishmaniasis & activity performed by military during outbreak | Sand fly species identified | Leishmania species identified in sand flies |
|---|---|---|---|---|---|---|---|---|
| Region | Country of deployment & Nationality of infected soldiers | | | | | | | |
| | Colombia [27,59] Colombian soldiers | NR | NR | NR | L. (V) braziliensis, L. panamensis, L. naiffi, L. lindenbergi, L. infantum, L. mexicana and L. lainsoni | CL Military operations in endemic areas within the country | NSF | NSF |
| | French Guiana French soldiers [16] German Soldiers [70] | > 4 | Wet | 30/858 (3%) [16] | L. (V) guyanensis | CL Military training course in the rainforest that involved keep watch during whole nights, sometimes laying on the ground or creeping in the underbrush. This military exercises also involves survival training in the jungle | NSF | NSF |
| | | NR | | | 22/1427 (2%) [16] | NR | | |
| | Panama Puerto Rican soldiers [20] US soldiers [71] | < 1 | Wet | 15/540 (3%) [20]; 3/60 (5%) [71] | L. (V) panamensis; L. (V) panamensis | CL Jungle warfare training in the Panama Canal Area that involves overnight sleeping at a mortar firing site | NSF | NSF |
| | Peru [21] Peruvian soldiers | 1 – 4 | Wet | 76/303 (25%) | L. (V) braziliensis and L. guyanensis | CL Short-term training in the Peruvian rainforest | NSF | NSF |
| | Suriname [31] Dutch soldiers | 1 – 4 | Wet | NR | L. naiffi | CL Military jungle training | NSF | NSF |
| | Southern United States (Texas, Kentucky & North Carolina) [57] US soldiers | NA | NA | NA | NA | NA | Lu. shannoni*, Lu. vexator, Lu. diabolica, Lu. anthophora, & Lu. aquilonia | NR |
| EMR | Afghanistan US soldiers [25,53] Dutch soldiers [34] British soldiers [33,72] German soldiers [61] | > 4 | Dry | 129/938 (14%) [34]; 20/120 (17%) [33] | L. major; L. major | CL Operation Enduring Freedom | Ph. alexandri, Ph. caucasicus, Ph. keshishiani, Ph. papatasi*, Ph. sergenti, Ph. turanicus, Se. clydei, Se. dreyfussi turkestanica, Se. grekovi & Se. murgabiensis | NR |
| | | NR | NR | NR | L. donovani complex | VL Operation Enduring Freedom | NSF | NSF |
| | Saudi Arabia [73] US soldiers | NR | NR | NR | L. tropica | VL and Asymptomatic Leishmania infections Operation Desert Storm | NSF | NSF |
| | Iraq Austrian soldiers [14] US soldiers [25,58,69] | NR | NR | NR 36/260 (13.8%) [14] 39/200 (19.5%) [69] | L. donovani/infantum complex | VL and Asymptomatic Leishmania infections NATO mission Kosovo Force (Austrian soldiers) Operation Iraqi Freedom (US soldiers) | Phlebotomus. Ph. papatasi*, Ph. sergenti, Ph. alexandri, & Sergentomyia spp | NR |

(Continued)

**Table 1.** (Continued)

| Deployment details | | | | CL cases/Total troops deployed (Cumulative incidence) | Leishmania species identified in cases | Type of leishmaniasis & activity performed by military during outbreak | Sand fly species identified | Leishmania species identified in sand flies |
|---|---|---|---|---|---|---|---|---|
| Region | Country of deployment & Nationality of infected soldiers | Duration (Months) | Season[a] | | | | | |
| | Oman [24] US soldiers | > 4 | Dry | NR | L. donovani/infantum complex | VL Operation Enduring Freedom | NSF | NSF |
| | Pakistan [64] Pakistani soldiers | NR | NR | NR | L. tropica, L. major and L. infantum | CL Soldiers serving in north and south Waziristan | NSF | NSF |
| | Syria [15] Austrian soldiers | 1 – 4 | Dry | 12/102 (12%) | L. donovani/infantum complex and L. tropica | Asymptomatic Leishmania infections UN or EU peacekeeping missions | NSF | NSF |
| | Lebanon [15] Austrian soldiers | 1 – 4 | Dry | 4/63 (6%) | None detected via PCR | Asymptomatic Leishmania infections UN or EU peacekeeping missions | NSF | NSF |
| EUR | BIH [15] Austrian soldiers | 1 – 4 | Dry | 2/61 (3%) | L. donovani/infantum complex | Asymptomatic Leishmania infections UN or EU peacekeeping missions | NSF | NSF |
| AFR | Egypt [13] Fijian soldiers US soldiers Australian soldiers Dutch soldiers Uruguayan soldiers British soldiers Colombian soldiers Italian soldiers | 1 – 4 | Dry | 9/60 (15%) 113/2582 (4%) | L. major | CL Multinational Force and Observers (MFO). An international peace keeping mission between Egypt and Israel. | Phlebotomus papatasi[*][b], & Sergentomyia antennata | L. major |
| SEAR | Sri Lanka [62] Sri Lankan soldier | NR | NR | NR | L. donovani | CL Soldiers serving at Mullaitivu and Kilinochchi districts | NSF | NSF |

Abbreviations: NA – Not applicable; NR – Not reported; NSF – No studies found; AMR – American region; EMR – Eastern Mediterranean region; EUR – European region; SEAR – South-East Asian region; AFR – African region; CL – cutaneous leishmaniasis; VL – visceral leishmaniasis; L – Leishmania; V – Viannia; BIH – Bosnia and Herzegovina.

Note: The nomenclature shown here is as reported in the citations and may not reflect current genera or subgenera nomenclature. The cumulative incidence is as reported for the respective countries based on the number of troops and number of leishmaniasis cases per deployment.

[a]Season is based on the reported months of deployment.

[b]Sand fly species with confirmed flagellate infections.

*Most abundant sand fly species reported in the study.

[57,60,63,65], two from EMR [58,61] and one from AFR [13]. Two studies confirmed sand fly infections with *Leishmania* and/or the specific *Leishmania* identities (Table 1). Among the four sand fly studies conducted in the AMR, three [57,60,63] employed morphological methods to identify the sand flies, but did not screen for *Leishmania*. One study [65] identified sand flies on morphological and molecular characteristics, and confirmed their infection with *L. (V.) braziliensis* by qPCR and restriction enzyme analysis. In the two studies conducted in EMR, one based sand fly identification on morphology [58], the other on morphology and molecular techniques [61], but neither study screened for *Leishmania* infection. The single study conducted in AFR [21] identified the sand flies morphologically, and sand fly infection with *L. major* on the basis of isoenzyme profiles. The sand fly species identities and those confirmed with *in situ* flagellate infections are listed in Table 1.

**Host immune responses to sand fly saliva exposure.** Host antigenic immune responses to proteins in sand fly saliva deposited when blood-feeding, usually measured by an enzyme-linked immunosorbent assay (ELISA) or Western blot analysis, are indicative of host exposure to sand fly bites. Four studies, all involving soldiers deployed to Iraq, measured their IgG antibody responses to sand fly whole salivary gland homogenate (SGH) antigens, two studies using *Phlebotomus (Ph.) papatasi* SGH [67,68] and two studies using *Ph. alexandri* SGH [66,69]). Antibody levels were significantly higher in troops post-deployment compared to pre-deployment, and significantly higher in those with parasitologically confirmed CL infections (cases) versus negative-CL diagnosed controls [67]. The salivary proteins in *Ph. papatasi* SGH frequently recognised were MW30 and MW64 [67], MW38 and 14 kDa [68]. Others included MW12/14, 15, 18, 26, 28, 32, 36, 42, 44, 46, and 52. Higher levels of IFN-γ, IL-6, IL-13, IL-10 and IL-17 cytokines were detected in soldiers exposed to *Ph. alexandri* SGH than in controls; highest levels were observed in SGH-positive asymptomatic soldiers[66]. Usually, these immune responses decline soon after repatriation to non-endemic regions, but with some exceptions [66].

## Discussion

### Symptomatic CL

CL is the most common form of leishmaniasis reported among deployed military personnel, similar to in civilian populations [1–3]. The meta-analysis of 12 estimates from eight studies showed a mean cumulative incidence of 10% (Fig 2), with a potential higher cumulative incidence in EMR countries (14%) compared to AFR (8%) or AMR (9%), though statistical differences between WHO regions were not detected. Notwithstanding, a high degree of heterogeneity (97.8%) was observed between estimates, as also detected between the seven estimates (four countries: Belize, French Guiana, Panama, and Peru) within the AMR region (97.9%) (Table D in S1 Text ). The possibility to pool estimates by aetiological agent(s) rather than region did not substantially alter the mean cumulative incidence value for symptomatic CL: *L. (V.) braziliensis* and *L. (L) mexicana*, being the most common, occur in AMR (Table 1) whereas the limited data excluded further pooling by aetiology. As such, the calculated incidence estimates presented here should be treated as preliminary indicators intended to lay a foundation for military health planning and to stimulate future generation of context-specific comparable data.

Heterogeneities within and between regions were not unexpected not least due to their geographical diversity, differences in aetiology and epidemiology, vector species' phenology, blood-feeding behaviour and vectorial competence, reservoirs, and timing of troop deployment relative to the climatic and environment conditions governing sand fly abundance [2,6,74–80]. In AMR countries, CL results from numerous *Leishmania* species, transmitted by

multiple known and putative vector species via a range of forest- and peridomestic-dwelling mammals (and possibly domestic animals) as hosts [39,81–84]. Current knowledge of the potential multiple transmission cycles in this region is poor. Human CL also occurs in the Old World due to *L. major* and *L. tropica* in EMR and AFR regions through better defined distinct zoonotic and anthroponotic cycles, respectively.

Sand fly abundance and biting intensities are generally climate-sensitive particularly to temperature and humidity [6,85–88]. Greater sand fly vector abundance is variably reported in dry or wet seasons and related to clinical CL incidence [78,86,89]. In the current study, the cumulative incidence of symptomatic CL was not statistically dissimilar between wet (10%) and dry (10%) seasons, though it should be acknowledged that temporal correlations between vector abundance and infection incidence are complex to interpret; the precise timing of inoculation is unknown, and infection prepatent periods are highly variable [90]. Indeed, "season" is a crude measure of the multiple climatic variables influencing sand fly population infection rates. Demography can also play a role: vector populations may be more infectious with increasing age, i.e., when the parous rate is highest- often at the end of the sand fly season [91]. For shorter durations of troop deployment, the timing of transmission may be more accurately attributed to a particular period or season. Table 1 suggests that leishmaniasis outbreaks primarily occur during the wet season in the New World, but during the dry season in the Old World. These trends are generally consistent with data reported in civilian populations [92], likely reflecting physiological and behavioural adaptations to local or regional environmental and climatic conditions [6,93–97]. For example, sand fly vectors in European states, tend to be most active in temperatures >15°C, but optimal temperature and humidity requirements for sand fly metabolism, growth and survival, and *Leishmania* development within the vector, vary considerably between species [6,98–100]. Heavy rainfall on the other hand, kills immature stages.

The analysis revealed differences in CL incidence with military deployment durations. The reason for the higher incidence in those deployed for 1–4 months compared to those lasting less than one month, or more than four months, is unclear. Speculatively, this may stem from heightened exposure during 1–4 months of deployment when personnel are less adapted to protective measures, in contrast to longer deployments giving more time to adhere to PPMs and make operational adjustments to reduce exposure. Future studies are needed to tease out such relationships.

### Asymptomatic *Leishmania* infections

The cumulative incidence of subclinical *Leishmania* infections was 11% across the studies, but again with significant inter-study heterogeneity (I2 = 78.47%). The identified infecting *Leishmania* species included the *L. donovani/infantum* complex and *L. tropica* among troops deployed to the EMR and EUR [15,69], with the highest incidence in Iraq (20%), compared to an incidence of <10% for *L. donovani/infantum* complex in Lebanon and BIH (Fig 4). Whilst *L. donovani* and *L. infantum* both typically cause VL, they differ in transmission patterns and geographical distribution. *Leishmania donovani*, found predominantly in the Indian subcontinent and East Africa [101,102], is anthroponotic, and transmitted by clinical cases as demonstrated by xenodiagnosis studies [103,104]. In contrast, VL due to *L. infantum*, is a zoonosis with domestic dogs as key reservoir, being prevalent in the Americas, Mediterranean countries, the Baltics, and parts of Central and East Asia [91,105–107]. Interestingly, some strains of *L. infantum* are associated with CL development in EUR, AMR and EMR [59,64,108]. Troops deployed to Syria and Bosnia reported lower incidences of infections with the *L. donovani/infantum* species complex compared to deployments to Iraq [109].

*Leishmania tropica*, which typically causes ACL or viscerotropic leishmaniasis, was less common in EUR deployments but highly endemic in EMR countries like Iraq, Syria, and

Lebanon [15]; infections with *L. tropica* was recorded in 2% (2/101) of Austrian troops deployed to Syria in 2013, likely reflecting the high transmission rates resulting in >50,000 civilian cases in the country in the previous year[15]. This highlights the potential increased risk to military personnel deployed to areas affected by mass population displacement and instability [110–115].

## Clinical VL and viscerotropic leishmaniasis

The number of documented military cases of VL were few relative to the reported cases of CL. The data on VL were primarily descriptive, lacking quantitative measures by which to conduct a similar meta-analysis as for CL.

All 13 VL cases had been deployed to the EMR [24,25,73], five to Iraq, Afghanistan and Oman, all of whom presented classical signs of VL. *Leishmania* amastigotes were detected in clinical samples in Iraq and Afghanistan [25]. The other eight cases were deployed to Saudi Arabia, and exhibited viscerotropic leishmaniasis, a mild form of VL usually associated with *L. tropica* infections, though the aetiology in these troops was not confirmed. Within Saudi Arabia, VL cases due to *L. donovani* are mainly restricted to the southwest of the country where *Ph. orientalis* is a likely vector, and the black rat (*Rattus ratus*) noted as a possible wildlife host [116], in addition to humans. Alternatively, the aetiology of the viscerotropic cases might have been *L. tropica*, transmitted anthroponotically, though usually causing ACL. In central and central west Saudi Arabia ACL and ZCL, due to *L. tropica* and *L. major* respectively, overlap [3,117]; key vectors likely include *Ph. sergenti and Ph. papatasi*, respectively. Wild rodent species, *Psammomys obesus* and *Meriones libycus*, are considered reservoir hosts of *L. major* in Central Asia, the Middle East and Africa [118–120].

## Diagnostics

Leishmaniasis diagnosis relies on clinical signs combined with confirmatory laboratory tests using parasitological, serological, and molecular techniques. Test selection depends on available facilities, resources and expertise [121]. In this review, 75% of studies used molecular techniques, with some incorporating parasitological/serological methods. Whilst parasitological methods remain the gold standard, it can be insensitive to detect infection, and requires invasive samples [104,122–131]. Molecular methods offer greater sensitivity but specificity for aetiological identification depends on the choice of gene targets [132–141]. Military personnel could benefit from isothermal platforms and Next-Generation Sequencing (NGS) for faster, cost-effective diagnostics with less invasive sampling [142–153].

Immunological diagnostics, including ELISA, IFAT, DAT and/or recombinant assays, were used in 7 studies to detect anti-*Leishmania* antibodies [152–158], although cross-reactivity between *Leishmania* species and with other pathogens (e.g., *Trypanosoma*, *Plasmodium*) can complicate interpretation [153,159–162]. Using recombinant *Leishmania* proteins, such as rK39 and rK28, can circumvent this issue [158,163–169]. Given slow decay rates of IgG antibody levels [170–172], distinguishing past exposure from active *Leishmania* infection requires clinical follow-up and robust controls; detectable antibodies are also influenced by the *Leishmania* aetiology, host immune status and stage of infection [121,127,160,173–177].

Tools for screening host exposure to sand fly saliva proteins deposited when blood-feeding are coming on-line[178–181], and were highlighted in four included studies. These reported host recognition of salivary proteins of *Ph. papatasi* (including MW30, MW64, MW38, 14 kDa) in troops post-deployment to Egypt, Jordan and Iraq [67,68]. Such anti-saliva antibody assays can provide valuable epidemiological insights into host-vector interactions [178,179,182–184], and serve as cost-effective tools for evaluating the impacts of vector

control interventions [185,186]. Statistical sample size requirements are likely to be lower, thus more feasible, than those to detect changes in *Leishmania* infection incidence.

### Personal protective measures

The use of different PPMs varied across studies. Meta-analysis of four studies of LLINs, comprising one cross-sectional study, one case-control study, and two cohort studies, suggested a uniform protective effect, reducing the risk of CL by nearly half amongst soldiers deployed to Afghanistan, French Guiana, Peru and Sri Lanka (Fig 5). This contrasts to the variable effectiveness of LLINs in civilian community trials [187–191] which may reflect differences in study designs. The propensity of vectors to blood-seek inside (endophagy) *versus* outside (exophagy) buildings/shelters is an important consideration regarding the relevance of ITNs/LLINs to deployed troops. Differences in sand fly host-seeking behaviour across endemic settings may partly explain discrepancies in the efficiency of PPMs, but there is a global need for sand fly behavioural studies to inform PPM best practise and vector control generally [192].

Permethrin-treated uniforms, or permethrin tablets for uniform re-treatment, are normally provided to deployed troops. Regular use of ITUs is shown to prevent vector bites, and lower both leishmaniasis and malaria case numbers in soldiers [193–200]. However, their effectiveness vary [194,201,202] by the degree of wear, number of washes, and exposure to chlorine [203–206]. ITUs covering arms and legs appeared to offer greater protection against CL risk than not being covered [21,34,207]. ITUs and long-sleeved clothing were associated with 28% and 78% protection against leishmaniasis, though the former was not statistically significant (Fig 5).

Meta-analysis of integrated PPMs, such as regular use of insect repellents and ITUs did not show statistically significant mean protective effects, whereas in one study, adoption of LLINs, insect repellent, and education was associated with zero leishmaniasis infections among German troops deployed to Afghanistan [208].

Consistent use of insect repellents has been shown to be low among soldiers despite availability [30]. Increased knowledge of leishmaniasis through troop education programmes was linked to improved repellent use among US soldiers, while non-compliance with PPMs was reported among younger Colombian recruits due to a lack of awareness [30,56]. Improved formulations of insect repellents and of educational programmes may increase compliance to PPMs [154].

Whether soldiers were deployed during peacetime for training, or during wartime operations, their risk of exposure to infected sand flies was commonly linked to similar activities including sleeping outside on the ground, nocturnal military drills (patrols and guarding in sentry positions), and extended periods outdoors. Development of CL or VL appears to be influenced more by the nature of the activities and environmental exposure than by the context of deployment (Table 1).

### Study limitations

The data included in this review were generated from a mix of cross-sectional, cohort and case-control study designs, and often without robustly selected or randomised control groups. Hence, the true effectiveness of adopted individual and multiple PPMs is largely not known and present results should be considered with caution. Likewise, leishmaniasis is not a single disease but a range of diseases along a clinical spectrum being associated with different aetiologies and transmission cycles including a diversity of vectors and hosts. Thus, generalisations based on the small number of heterogeneous studies and their geographical and epidemiological conditions, complicates interpretation. The raw data extracted from included studies

were transformed where possible to increase the number of studies that could be included in the meta-analysis, and the robustness and reliability were maximised by testing different effect sizes before using the Freeman-Tukey transformed proportions. Nonetheless, the small number of studies available for each meta-analysis prevented exploring sources of heterogeneity using meta-regression and from conducting a sensitivity analysis as initially intended.

In sum, future research should prioritise well-designed intervention studies that evaluate the effectiveness of PPMs, both individually and in combination, across diverse deployment settings, incorporating robust experimental designs with well-defined control groups and sufficient replication to meet statistical requirements. Cross-sectional and case-control studies can identify associations but do not demonstrate (mechanistic) cause and effect. Additionally, it is essential to improve health education programs pre-deployment to raise awareness of leishmaniasis and other vector-borne diseases, emphasising mitigation strategies and correct use of PPMs. Practical innovative tools and technologies are ultimately needed to enhance vector control and disease prevention in military field settings.

## Supporting information

**S1 Text. The supporting information file contains all the relevant data including the protocol underlying this systematic review and meta-analysis.**
(DOCX)

## Acknowledgments

We would like to thank the University of Warwick library team, who supported the sourcing of references for this review. We would also like to thank the military departments that helped us access military literature.

## Author contributions

**Conceptualization:** Ngwa Niba Rawlings, Mark Bailey, Orin Courtenay.

**Data curation:** Ngwa Niba Rawlings.

**Formal analysis:** Ngwa Niba Rawlings.

**Funding acquisition:** Mark Bailey.

**Investigation:** Ngwa Niba Rawlings, Mark Bailey, Orin Courtenay.

**Methodology:** Ngwa Niba Rawlings, Mark Bailey, Orin Courtenay.

**Project administration:** Ngwa Niba Rawlings, Mark Bailey, Orin Courtenay.

**Resources:** Mark Bailey, Orin Courtenay.

**Software:** Ngwa Niba Rawlings.

**Supervision:** Mark Bailey, Orin Courtenay.

**Validation:** Mark Bailey, Orin Courtenay.

**Visualization:** Orin Courtenay.

**Writing – original draft:** Ngwa Niba Rawlings.

**Writing – review & editing:** Ngwa Niba Rawlings, Mark Bailey, Orin Courtenay.

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
