## [Decision Letter · Decision Letter 0]

3 Jan 2025

PNTD-D-24-01652Leishmaniasis in deployed military populations: A systematic review and meta-analysisPLOS Neglected Tropical DiseasesDear Dr. Niba Rawlings, Thank you for submitting your manuscript to PLOS Neglected Tropical Diseases. After careful consideration, we feel that it has merit but does not fully meet PLOS Neglected Tropical Diseases's publication criteria as it currently stands. Therefore, we invite you to submit a revised version of the manuscript that addresses the points raised during the review process. Please submit your revised manuscript within 30 days Mar 04 2025 11:59PM. If you will need more time than this to complete your revisions, please reply to this message or contact the journal office at plosntds@plos.org. Please include the following items when submitting your revised manuscript: * A rebuttal letter that responds to each point raised by the editor and reviewer(s). You should upload this letter as a separate file labeled 'Response to Reviewers '. This file does not need to include responses to any formatting updates and technical items listed in the 'Journal Requirements' section below. * A marked-up copy of your manuscript that highlights changes made to the original version. You should upload this as a separate file labeled 'Revised Manuscript with Track Changes '. * An unmarked version of your revised paper without tracked changes. You should upload this as a separate file labeled 'Manuscript '. If you would like to make changes to your financial disclosure, competing interests statement, or data availability statement, please make these updates within the submission form at the time of resubmission. Guidelines for resubmitting your figure files are available below the reviewer comments at the end of this letter. We look forward to receiving your revised manuscript. 

Kind regards,

Richard Reithinger

Academic Editor

Susan Madison-AntenucciSection EditorPLOS Neglected Tropical Diseases

Shaden Kamhawi

co-Editor-in-Chief

Paul Brindley

co-Editor-in-Chief

**Journal Requirements:**

At this stage, the following Authors/Authors require contributions: Ngwa Niba Rawlings, Mark Bailey, and Orin Courtenay. Please ensure that the full contributions of each author are acknowledged in the "Add/Edit/Remove Authors" section of our submission form.

2) Please amend your detailed Financial Disclosure statement. This is published with the article. It must therefore be completed in full sentences and contain the exact wording you wish to be published.

**Reviewers' comments:** Reviewer's Responses to Questions

**Key Review Criteria Required for Acceptance?**

**Methods**

-Are the objectives of the study clearly articulated with a clear testable hypothesis stated?

-Is the study design appropriate to address the stated objectives?

-Is the population clearly described and appropriate for the hypothesis being tested?

-Is the sample size sufficient to ensure adequate power to address the hypothesis being tested?

-Were correct statistical analysis used to support conclusions?

-Are there concerns about ethical or regulatory requirements being met?

Reviewer #1: The review selection criteria for inclusion/exclusion of literature could be improved:

Search strategy and selection criteria: The text states, “Specific subtopics or aspects of leishmaniasis in the military were considered, such as…” How were the subtopics “incidence” (not cumulative) and “prevalence” used in the search and selection criteria? Since cumulative incidence (or attack rate), characterization of sand fly and Leishmania species, sand fly saliva exposure, effectiveness of PPMs, and associated risk are listed as inclusion criteria, but not “incidence” and “prevalence2, I recommend removing "incidence" and "prevalence" from this sentence to avoid confusion.

As described below, the meta-analysis carried out with few studies raises concerns when drawing conclusions.

Reviewer #2: The methods are strong and well explained. The systematic review and the meta analysis are thoroughly detailed.

**Results**

-Does the analysis presented match the analysis plan?

-Are the results clearly and completely presented?

-Are the figures (Tables, Images) of sufficient quality for clarity?

Reviewer #1: Cumulative incidences of cutaneous leishmaniasis: Pooling incidence data from geographically disparate regions with differing reservoirs, vectors, and parasites raises questions about its validity. Even within the same WHO region, heterogeneity can be extremely high. Conducting a meta-analysis with only three studies also raises concerns—is it valid? Given the high inter-study heterogeneity, what is the rationale behind generating a unified indicator for such disparate data?

Adoption of PPMs and associated risk of developing leishmaniasis: Replace “however the latter two PPMs were not individually significant from controls” with “however the latter two PPMs were not individually significantly different from controls.”

Additionally, please specify which PPMs were used in the studies. Since the PPMs can vary significantly, the effectiveness and associated risk of developing leishmaniasis may depend heavily on the type of PPMs employed.

Symptomatic visceral leishmaniasis (VL): The meta-analysis for cutaneous leishmaniasis (CL) was based on cumulative incidence and odds ratios for developing leishmaniasis associated with PPMs. However, the description of the studies on VL does not address these aspects. Is this because such data were not available in the literature? Why was a meta-analysis of cumulative incidence possible for asymptomatic CL but not for VL, given that both analyses involved three studies? Please clarify.

Reviewer #2: The results are very clearly exposed. A few comments:

- page 9: one study reported both CL and VL: please indicate here in which country, as I do not see this study in the Table

-page 11: the duration of deployment lasting 1-4 montsh appeared to result in a higher cumulative incidence comapred to < 1 month (this is understandable) and > 4 months (how do the authors explain this? a duration > 4 months should be associated wiht a higher risk.

- page 11: screening for Leishmania DNA by PCR : please specifiy which kind of samples were used, as this can have an important influence on sensitivity (swabs, skin biopsy, scrapings...)

- page 12: the abbreviation LLIN has not been explained

**Conclusions**

-Are the conclusions supported by the data presented?

-Are the limitations of analysis clearly described?

-Do the authors discuss how these data can be helpful to advance our understanding of the topic under study?

-Is public health relevance addressed?

Reviewer #1: Discussion

Symptomatic CL: The first two sentences in this section are unnecessary. You should begin by emphasizing the key findings of your review and meta-analysis.

Again, what is the value of generating a unified indicator for certain metrics when heterogeneity is so high? Please address this in the discussion section. Beyond stating the study's limitations, assess how reliable the incidence values are, given the elevated heterogeneity. While this approach may facilitate comparisons between studies, it seems less effective for synthesizing the available scientific information when contexts differ so greatly. You might consider drawing conclusions about this issue.

The higher cumulative incidence (20% [95% CI: 16, 25]) associated with deployments lasting 1–4 months compared to those lasting less than 1 month or more than 4 months (<7%) was not discussed. Please address this finding.

Reviewer #2: (No Response)

**Editorial and Data Presentation Modifications?**

Reviewer #1: Introduction

Some sentences lack citations, e.g., the last sentences of paragraphs 1 and 2 in the Introduction.

You use "PPM" to refer to "personal protective measures" in the summary but "personal preventative measures" in the Introduction. Please unify the terminology throughout the manuscript.

Discussion

Spell out the full names of species when starting a sentence, e.g., “Leishmania donovani, found predominantly in the Indian subcontinent and East Africa…”

Reviewer #2: (No Response)

**Summary and General Comments**

Reviewer #1: This review demonstrates a thorough and rigorous effort in searching and performing a meta-analysis of the literature on leishmaniasis in deployed military populations. However, the limited number of studies included in the meta-analysis prevents robust conclusions from being drawn based on synthesized information, such as the cumulative incidence.

Reviewer #2: This is a very interesting study. The question of leishmaniasis outbreaks among military personnel is a very important issue and this paper brings clarity to the matter.

The non-inclusion of papers in languages other than English is a real weakness as this Neglected Tropical Diseases can hardly be fathomed without considering works in French, Spanish and Portuguese for the New World, and Middle East languages for the Old World.

Besides, using papers only in English narrows the scope by focusing on US and UK military populations, when other armies with limited resources could be associated with different results.

I think Table 1 should indicate, for each study, the nationality of infected soldiers (when available). The conclusions in terms of immunity and prevention are very different if the infection occurs in Saudi soldiers living in endemic areas (for example) or US soldiers deployed for a short term in Saudi Arabia.

If the authors found no study involving local military populations infected in their own country, this point should be discussed. For example, it is known that outbreaks occur in French-speaking African armies in Western Africa, but these outbreaks are seldom published (and usually in French).

Another point that should be discussed (if possible) is the type of activity performed by the military at the time of each outbreak. This should be detailed in Table 1 and discussed. For example, the outbreak in French Guiana in 2020 was linked to intense training in the rainforest and not actual fighting, which can hardly be compared with real frontline situations of US troops in the Middle East.

Page 23, the authors mention the importance of vectors endophagy/exophagy. There are differences in vectors behaviours between each endemic area (due to phlebotomines local ecology). Can these differences explain the discrepancies in the efficiency of PPMs?

Finally, the authors rightly point the difficult definition of rain or dry season across a large range of heterogenous countries and studies. However, it seems to me that Table 1 suggests that outbreaks mostly occur during wet seasons in the New World and dry season in the Old World, is that in line with usual data in civilian populations?

PLOS authors have the option to publish the peer review history of their article (what does this mean? ). If published, this will include your full peer review and any attached files.

**Do you want your identity to be public for this peer review?** For information about this choice, including consent withdrawal, please see our Privacy Policy .

Reviewer #1: No

Reviewer #2: **Yes: ** Romain Blaizot

**Figure resubmission:** While revising your submission, please upload your figure files to the Preflight Analysis and Conversion Engine (PACE) digital diagnostic tool, https://pacev2.apexcovantage.com/. PACE helps ensure that figures meet PLOS requirements. To use PACE, you must first register as a user. Registration is free. Then, login and navigate to the UPLOAD tab, where you will find detailed instructions on how to use the tool. If you encounter any issues or have any questions when using PACE, please email PLOS at figures@plos.org. Please note that Supporting Information files do not need this step. If there are other versions of figure files still present in your submission file inventory at resubmission, please replace them with the PACE-processed versions.**Reproducibility:** To enhance the reproducibility of your results, we recommend that authors of applicable studies deposit laboratory protocols in protocols.io, where a protocol can be assigned its own identifier (DOI) such that it can be cited independently in the future. Additionally, PLOS ONE offers an option to publish peer-reviewed clinical study protocols. Read more information on sharing protocols at https://plos.org/protocols?utm_medium=editorial-email&utm_source=authorletters&utm_campaign=protocols

---

## [Editor Report · Decision Letter 1]

10 Feb 2025

Dear Mr. Niba Rawlings,

We are pleased to inform you that your manuscript 'Leishmaniasis in deployed military populations: A systematic review and meta-analysis' has been provisionally accepted for publication in PLOS Neglected Tropical Diseases.

Best regards,

Richard Reithinger

Academic Editor

Susan Madison-Antenucci

Section Editor

Shaden Kamhawi

co-Editor-in-Chief

Paul Brindley

co-Editor-in-Chief

Thank you for addressing all of the reviewers' concerns.

---

## [Editor Report · Acceptance letter]

Dear Mr. Niba Rawlings,

We are delighted to inform you that your manuscript, "Leishmaniasis in deployed military populations: A systematic review and meta-analysis," has been formally accepted for publication in PLOS Neglected Tropical Diseases.

Best regards,

Shaden Kamhawi

co-Editor-in-Chief

Paul Brindley

co-Editor-in-Chief
